# Secondary Mechanisms of Neurotrauma: A Closer Look at the Evidence

Sina Aghili-Mehrizi *, Eric Williams, Sandra Yan, Matthew Willman, Jonathan Willman and Brandon Lucke-Wold *

Department of Neurosurgery, University of Florida, Gainesville, FL 32608, USA; williamseric@ufl.edu (E.W.); sandra.yan@neurosurgery.ufl.edu (S.Y.); matthewwillman@ufl.edu (M.W.); jonathanwillman@ufl.edu (J.W.)
* Correspondence: aghilimehrizi.s@ufl.edu (S.A.-M.); brandon.lucke-wold@neurosurgery.ufl.edu (B.L.-W.)

**Abstract:** Traumatic central nervous system injury is a leading cause of neurological injury worldwide. While initial neuroresuscitative efforts are focused on ameliorating the effects of primary injury through patient stabilization, secondary injury in neurotrauma is a potential cause of cell death, oxidative stress, and neuroinflammation. These secondary injuries lack defined therapy. The major causes of secondary injury in neurotrauma include endoplasmic reticular stress, mitochondrial dysfunction, and the buildup of reactive oxygen or nitrogenous species. Stress to the endoplasmic reticulum in neurotrauma results in the overactivation of the unfolded protein response with subsequent cell apoptosis. Mitochondrial dysfunction can lead to the release of caspases and the buildup of reactive oxygen species; several characteristics make the central nervous system particularly susceptible to oxidative damage. Together, endoplasmic reticulum, mitochondrial, and oxidative stress can have detrimental consequences, beginning moments and lasting days to months after the primary injury. Understanding these causative pathways has led to the proposal of various potential treatment options.

**Keywords:** neural injury; oxidative stress; endoplasmic reticulum stress; apoptosis; mitochondrial dysfunction





## 1. Introduction

Neurotrauma is defined as an external force causing alterations in central nervous system (CNS) functioning or evidence of new CNS pathology [1–3]. It is a worldwide leading cause of morbidity and mortality in both the young and elderly, with traumatic brain injury (TBI) in the United States alone impacting 2.8 million individuals per year [1,3–5]. Initial neuroresuscitation focuses on treating the primary injury, such as basic life support, maintaining cerebral perfusion, seizure prophylaxis, and surgery [5,6]. Despite the substantial improvements in these resuscitative efforts following traumatic CNS injuries, one of the inevitable sequelae of neurotrauma is secondary injury, or the molecular and chemical response to primary injury. Direct impact can cause the transfer of force dynamics through the dura, cerebrospinal fluid, and brain parenchyma. This can result in the shearing of axons, the disruption of cell membranes, and fluid shifts, resulting in edema. This primary injury transitions to the secondary injury response over time.

Secondary injury in neurotrauma has been linked to endoplasmic reticulum (ER) and mitochondrial dysfunction as well as oxidative stress [7–10]. Stress to these organelles triggers a cascade of events which ultimately activate neuroinflammatory pathways [7,11–13]. Signs of neuroinflammation, or the CNS's immune response, begin to develop shortly after the primary injury [6,14,15]. Some overt changes include ventricular enlargement, edema, white matter atrophy, or gray matter atrophy, while microscopic apoptosis, autophagy, axonal injury, and necrosis have all been visualized [7,15–23]. Secondary injury can persist for days to months following neurotrauma, with many potentially life-long implications,

oftentimes more taxing than the primary insult [14,15]. Namely, acute and chronic neuroinflammation have been linked with initiating or exacerbating the progression of neurodegenerative diseases and processes such as Alzheimer's or multiple sclerosis [11,12,24,25]. To date, there is no FDA-approved medication for the prevention or treatment of secondary injury in neurotrauma [26]. Thus, the importance of understanding and addressing the pathomechanisms of secondary injury cannot be understated. The goal of the present study is to review the impact of neurotrauma at cellular and molecular levels to identify potential treatment modalities to secondary injury.

## 2. ER Stress

The ER is a continuous, membrane-enclosed series of flattened sacs within the cytoplasm of eukaryotic cells. This organelle is essential for multiple cellular functions such as lipid biosynthesis, calcium cation storage, post-translation modifications, protein folding, and nascent protein transport [27–34]. It houses transmembrane proteins while also synthesizing, folding, and secreting most extracellular proteins [27]. The ER's relatively high calcium concentration is crucial to maintaining the electrochemical environment necessary to perform these roles [8]. Cell stressors such as hypoxia, starvation, trauma, and infection alter the ER's environment and thus, the folding and sorting of proteins within the ER. This leads to an accumulation of unfolded proteins [8,33–36]. When unfolded proteins reach a critical threshold, the ER is said to be under stress, and the ER's unfolded protein response (UPR) pathway is initiated [27–33].

The UPR (Figure 1) is an evolutionarily conserved signal cascade that works to restore protein homeostasis by reducing the number of unfolded proteins through protein expression alterations [28,32–34,37–39]. The UPR functions through three ER transmembrane protein sensors: inositol-requiring kinase 1 (IRE1Aα), pancreatic ER eIF2 kinase (PERK), and activating transcription factor 6 (ATF6) [30–33,37]. Under stress, ATF6 is cleaved by proteases to produce a transcriptionally active polypeptide that translocates to the nucleus, where it upregulates various proteins such as chaperones [27,33,35,37,40]. The activation of PERK and IRE1α is mediated by binding of their luminal N-terminal sequences by accumulated unfolded proteins [35]. These three sensors interact with an ER chaperone protein glucose-regulated protein 78 (GRP78), which is a central regulator and marker for ER stress [41]. GRP78 disassociates from the UPR signal sensors mediating the intracellular signaling pathways involved in the UPR [41]. Utilizing the UPR under acute stress can restore ER (and thus cellular) homeostasis. However, prolonged or high stress states can result in UPR-activated cell death via apoptosis [29,33–35,37,38,40]. In addition to UPR activation, the accumulation of misfolded protein within the ER is known to affect the ubiquitin proteasome system (UPS). Under normal conditions, the UPS is responsible for degrading proteins targeted for destruction. The disruption of this mechanism results in the further accumulation of protein and protein aggregates, worsening ER stress and contributing to disease processes [40,42,43].

Neurotrauma is a relatively common cause of neuronal ER dysfunction via oxidative stress, inflammation, and metabolic disturbance [8,13,44]. Several studies show elevation in UPR stress markers following TBI and spinal cord injury (SCI), which has also been evidenced in a variety of neurodegenerative diseases [28,30,34,36,38–40,42–50]. ER stress-related protein aggregates are present in HD, ALS, and PD, while upregulated IRE1α has been evidenced in AD, PD, and ALS [8,51–54]. Likewise, PERK hyperactivation has been evidenced in progressive supranuclear palsy (PSP) [8,51]. Interestingly, UPR upregulation occurs prior to symptom onset in these diseases, and one study showed that the suppression of UPR signals may alleviate AD-related memory deficit [51]. These findings suggest neurotrauma-induced ER stress may lead to pathological findings similar to those seen in familiar neurodegenerative diseases.

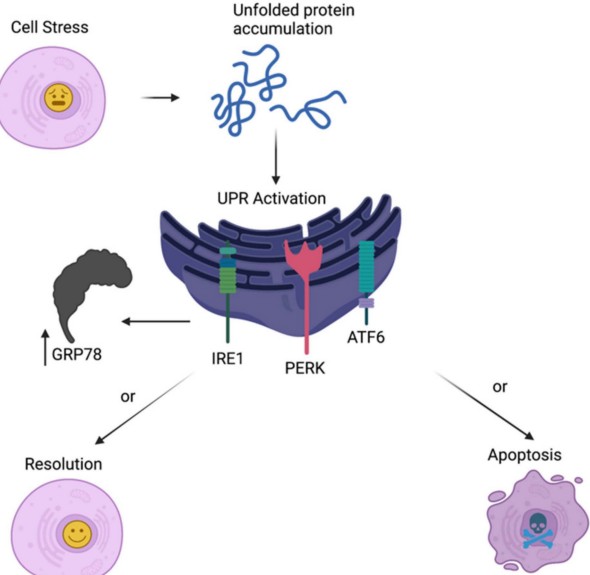

**Figure 1.** Unfolded protein response cascade. Cell stressors such as traumatic injury lead to accumulation of unfolded proteins in the ER, which activates the UPR transmembrane protein sensors IRE1A$\alpha$, PERK, and ATF6. These protein sensors lead to upregulation of GRP78, a signal for ER stress. Additionally, activation of the UPR transmembrane proteins results in either restoration of homeostasis or apoptosis through downstream mechanisms. Created with BioRender.com.

## 3. Mitochondrial Dysfunction

Mitochondria play a crucial role in ATP production, allowing proper cellular function and repair, $Ca^{2+}$ buffering, apoptosis, and the regulation of reactive oxygen species (ROS) in the cell [55,56]. In the setting of neurotrauma, reduced blood supply, and thus oxygen, inhibits aerobic metabolism through the mitochondrial electron transport chain (ETC), substantially reducing ATP production [57]. This forces mitochondria to upregulate anaerobic metabolism through lactic acid production to satisfy cellular energy requirements [58]. In addition to decreased ATP production, extracellular lactic acid accumulation is a strong indicator of mitochondrial dysfunction and correlates with worse outcomes in TBI patients [59]. Although oxygen depletion secondary to neurotrauma plays a critical role in decreased mitochondrial ATP production, the restoration of oxygen supply to tissue affected by neurotrauma alone may not be sufficient to restore adequate mitochondrial ATP production.

Mitochondrial dysfunction can occur in the setting of neurotrauma without any apparent indication of ischemia through the unregulated accumulation of $Ca^{2+}$ within the cytoplasm and mitochondria, causing excitotoxicity [60–62]. Neurotrauma-induced excitotoxicity occurs via the stimulation of glutamate (NMDA) receptors, resulting in the opening of voltage-gated $Ca^{2+}$ channels and the uptake of $Ca^{2+}$ (Figure 2) [63,64]. With an influx of $Ca^{2+}$ and a loss of $Ca^{2+}$ homeostasis, mitochondria $Ca^{2+}$-dependent proteases and phospholipases are activated, upregulating the production of ROS [26,65]. This neurotrauma-induced elevation of mitochondrial ROS drives oxidative stress in neurons, discussed in Section 4 [66]. Furthermore, the excessive accumulation of $Ca^{2+}$ can induce apoptotic cell death due to mitochondrial membrane compromise, as seen in outer membrane permeabilization (MOMP) and the formation of mitochondrial permeability transition pores (mPTP) [67,68]. With the disruption of the mitochondrial membrane, mPTP and MOMP cause the release of cytochrome c (cyt c) and other mitochondrial proteins into the cytoplasm of the cell [69,70]. These mitochondrial proteins activate caspases, namely caspase-3, resulting in caspase-dependent cell death [71]. However, mitochondrial membrane compromise alone can cause caspase-independent cell death as well [72]. Due to its catastrophic affects to the cell, preventing $Ca^{2+}$ dysregulation and consequent mPTP progression is a

major focus of therapeutic investigations to limit cellular apoptosis and preserve the ability of the mitochondria to produce energy [69].

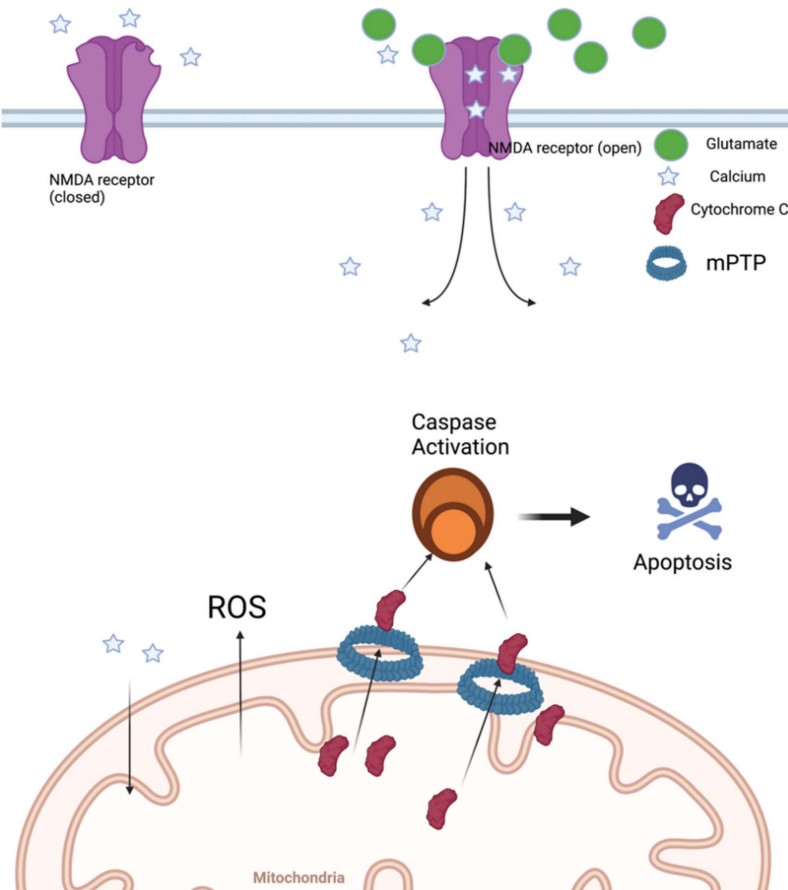

**Figure 2.** Neurotrauma-induced excitotoxicity of glutamate NMDA receptors allows for influx of $Ca^{2+}$ and overloads mitochondrial $Ca^{2+}$ homeostasis. The resulting instability of the mitochondrial membrane causes mitochondrial proteins such as cytochrome c to spill out of the mitochondria through mPTP. Caspases can be activated by these proteins, ultimately inducing apoptosis. ROS production is upregulated, as excessive $Ca^{2+}$ promotes ROS production through the activation of $Ca^{2+}$-dependent proteases and phospholipases. Reprinted/adapted with permission from Ref. [73]. 2022, BioRender.

## 4. Oxidative Stress

Free radicals are atoms, molecules, or ions with unpaired electrons that are formed via covalent bond disruption. These compounds are chemically unstable, causing them to react with either other free radicals or nonradical molecules [74]. Under physiologic conditions, these reactive chemical species can be produced via oxidative phosphorylation, the biotransformation of proteins in the ER, or enzymatic reactions [74]. One major producer of ROS is NADPH oxidase (NOX). The body contains numerous antioxidants in the form of enzymes such as superoxide dismutase (SOD), glutathione peroxidase, or catalase, which prevent serious harm from these reactive species [75]. The expression of these enzymes is controlled by transcription factors such as Nrf2, which binds to antioxidant response elements (AREs) to induce the transcription of detoxifying and antioxidant genes [76].

Secondary injury in neurotrauma through various cascades, including those discussed in the above sections of this review, contribute to excessive free radical formation, further exacerbating injury. The buildup of these reactive species overwhelms the antioxidant response, creating a deadly cycle of continuous free radical formation [75,77]. These oxidative species go on to interact with proteins, lipids, carbohydrates, and nucleic acids, leading

to irreversible cellular damage termed "oxidative stress" or "oxidative damage." [26,77]. The CNS is particularly sensitive to oxidative damage because of its relatively high lipid concentration and abundant oxidative metabolism. Animal models of TBI and SCI show glutamate-induced excitotoxicity via NMDA receptors, resulting in elevations of intracellular calcium with subsequent elevations in superoxide ($O_2\bullet^-$) production via NOX moments after injury [77,78]. With the CNS's high lipid levels, superoxide is able to readily induce lipid peroxidation (LP). Among its numerous harmful effects, LP results in the leakage of lysosomal hydrolytic enzymes and $Ca^{2+}$ from the mitochondria, ultimately causing apoptosis through the aforementioned mechanisms [74]. Superoxide can also react with local endothelial cell nitric oxide (NO), forming peroxynitrite [79]. Peroxynitrite has been implicated in blood–brain barrier (BBB) leakage, neuroinflammation, edema, and mitochondrial dysfunction. Its effects on the BBB allow cells of the immune system to enter the CNS, further exacerbating disease [74,80].

## 5. Emerging Treatments

### 5.1. ER Stress

Potential treatment options targeting the UPR pathway look to ameliorate ER stress as a cause of secondary injury in neurotrauma. Specifically, two drugs acting on eIF2$\alpha$ phosphorylation have shown promising results in recent studies on animal models. Salubrinal, an eIF2$\alpha$ dephoshorylation inhibitor, has recently been shown to decrease ER-stress-associated neuronal cell death via disrupting caspase-3-mediated apoptosis and neuroinflammation after TBI [81–84]. Similarly, Guanabenz and its derivatives (e.g., sephin1) have been shown to increase eIF2$\alpha$ phosphorylation [8,85]. Recent studies examining the therapeutic effect of Guanabenz and sephin1 have shown reductions in unfolded protein production, ER stress, and TBI neural deficits [86–90]. Additionally, Tauroursodeoxycholic acid (TUDCA), an endogenous bile acid, is another potential treatment targeting ER stress. Previous studies have shown TUDCA's ability to promote blood vessel repair, reduce arterial stiffness, and decrease endothelial dysfunction in rodent models of type 2 diabetes [91,92]. However, recently, TUDCA use in rodent models of subarachnoid hemorrhage has been shown to increase cerebrovascular perfusion, decrease GRP78 expression, and inhibit PERK, eIF2$\alpha$, and ATF4 signaling, ultimately decreasing ER-stress-mediated apoptosis [41].

### 5.2. Mitochondrial Dysfunction

A prominent cause of mitochondrial stress (and thus increased ROS formation) in neurotrauma is calcium overload via glutamate–NMDA interaction. While preliminary research focused on the broad-stroke downregulation of the NMDA receptor has proven to be counterproductive with many side effects and a limited window of therapy, research has shown that there are two NMDA receptors of interest: synaptic NMDA receptors which increase nuclear $Ca^{2+}$ and antioxidant production and extra-synaptic NMDA receptors which promote cytoplasmic $Ca^{2+}$ and mitochondrial stress [93]. Recent research has focused on the selective inhibition of extra-synaptic NMDA receptors via memantine, a well-studied neuroprotective drug in AD [80,94]. Preliminary studies in rodent models have shown that the memantine-mediated downregulation of extra-synaptic NMDA receptors in the setting of TBI is protective against mitochondrial stress and neuronal damage [80,94].

Another treatment option in mitochondrial dysfunction looks to inhibit mPTP formation by reproducing the effects of cyclosporin A (CsA). CsA has been well-documented in inhibiting apoptotic cell death in various cells, including neurons, presumably through its inhibition of the release of pro-apoptotic factors by mPTP [95–97]. However, its cytotoxic effects have limited CsA as a potential treatment option in neurotrauma [98,99]. NIM811, a cyclosporin A (CsA) analog, is a less toxic alternative currently under investigation, primarily for SCI [98]. In addition to preserving mitochondrial function, this potential treatment has been shown to promote tissue sparing and functional recovery in rodent models of SCI [98].

### 5.3. Antioxidant Therapy

Reactive species production is one of the more well-studied mechanisms of secondary injury in neurotrauma, and thus, a broader variety of potential treatment options targeting various pathways in their production and removal are currently under investigation. Edaravone is a multi-target compound that has been used in Japan since 2001 for its scavenging of free radicals post-ischemic stroke [26]. Recently, it was approved by the FDA for ALS treatment because of its ability to increase antioxidant enzyme expression and to prevent cyt c and caspase-3 release in the mitochondria [26]. Despite its use in ALS and stroke, there is limited studies on its safety and efficacy in TBI patients. In rodent models of TBI, edaravone has been shown to significantly reduce apoptotic activity in a dose-dependent fashion, with one study showing its benefits when administered up to 6 h following controlled cortical impact (CCI) [100]. Several other studies have shown decreased evidence of LP following edaravone administration as well as increased Nrf2 expression [100–102]. Another potential therapy that has shown promising results in rodent models of TBI is Apocynin/TBHQ. Apocynin, a NOX inhibitor, and TBHQ, a NRF2 activator, when used as a dual-blend therapy, can salvage both white and gray matter when administered up to 2 h after TBI [103]. Furthermore, Mitoquinone (MitoQ) is being investigated as an antioxidant that targets the mitochondrial ETC. Its actions on the mitochondria lead to a series of downstream effects that ultimately increase Nrf2 release and thus antioxidant enzyme gene expression [26,76]. Although its effects in PD, HD, AD, and ALS have been widely studied, the investigation of its benefits in TBI has only recently begun [76,104–108].

### 5.4. Immunoglobulin

Antibodies are a broad field of therapies that have garnered interest in the treatment of TBI partly due to their theoretically targeted nature. Kondo et al. demonstrated that TBI in mice induced cis phosphorylates-tau (p-tau) production, axonal interference, mitochondrial dysregulation, and subsequent apoptosis in a process they labeled "cistauosis" [109]. In addition, Kondo et al. showed that an anti-cis p-tau-specific antibody could rescue the majority of cistauosis-induced consequences, including apoptosis and mitochondrial dys-function [109]. The concept that tau pathology is linked to mitochondrial dysregulation has been endorsed by studies from the field of Alzheimer's research [110–113]. Kondo et al.'s findings and the possible use of a p-tau therapeutic antibody were subsequently supported by a number of recent studies [114–116]. One study of note demonstrated a statistically significant negative correlation between Glasgow Coma Scale results and cis p-tau levels in the CSF of human TBI patients [117]. This further endorses the notion that cis p-tau is directly associated with worse TBI results and that cis p-tau antibodies may have therapeutic value.

Another potential target of immunoglobulin therapy in TBI is the molecule caveolin. Increased caveolin-1 levels in the CSF have been associated with worse outcomes in TBI [118]. In addition, caveolin-1 mouse knockout was correlated with decreased inflammation and oxidative stress in the setting of TBI [119]. Caveolin-3, found largely in astrocytes within the CNS, is linked with a reduction in endothelial nitric oxide synthase (eNOS) [120–122]. This may promote oxidative injury, given the positive association between eNOS and reduced oxidative stress [123,124]. Further research in the field of caveolin modulation is vital before therapies may be developed.

### 5.5. Cell-Based Therapy

Stem-cell-based therapy for traumatic brain injury (TBI) has been a topic of research for many years and remains one of the foremost options as a future therapeutic. The divisions of stem cells used in TBI research include neural stem cells (NSCs), mesenchymal stem cells (MSCs), endothelial progenitor cells (EPCs), and multipotent adult progenitor cells (MAPCs) [125]. In recent years, research has focused more on the use of MSCs. MSCs have been shown to migrate to the cite of TBI, inhibit microglia activation and peripheral

leukocyte migration, inhibit proinflammatory cytokines and oxidative stress, and repair injured tissue through the upregulation of growth factors (e.g., VEGF) and neurotrophic factor transcription (e.g., BDNF and GDNF) [125–128]. In addition, there is new evidence that MSCs may increase ATP production in the setting of ischemia through a process known as mitochondrial transfer, in which mitochondria are transferred from the MSC to local cells through a novel exocytotic process [129,130]. Two concerns regarding stem cell therapy in TBI include potential tumorgenicity and embolism formation [131,132]. While studies have repeatedly shown the increased risk of embolism formation in high-dose stem cell therapy, data have been inconclusive concerning the enhanced probability of tumorgenicity, with the latest studies finding no heightened risk [132,133].

*5.6. MSC-Exosomes*

In recent years, an innovative and focused application of TBI stem cell therapy called MSC-derived exosomes (MSC-exosomes) has emerged as a promising new therapy. Almost every cell in the human body exudes extracellular vesicles. There are two major categories of extracellular vesicles—ectosomes and exosomes, which are comparatively smaller with an average diameter of 100 nm [134]. MSC-exosomes contain many of the products of their parent MSC cells, including nucleic acids, lipids, and proteins [134,135]. In addition, research has shown that many of the benefits of MSCs are not based on the stem cells' ability to differentiate and replace dead tissue, but rather on their ability upregulate growth factors and anti-inflammatory mediators that reduce oxidative stress and mitochondrial damage through exosome production and modulatory signaling [135–138]. Consequently, MSC-exosomes may offer many of the same advantages as MSCs without the cell-based risk factors [135,139]. Recent research has shown that MSC-exosomes may upregulate AKT and ERK pathways and counteract the effects of ER-stress-induced apoptosis while simultaneously downregulating genes associated with ER stress [140–142]. One study by Zhang et al. found that TBI rats treated with MSC-exosomes showed the significant rescue of neurological deficits, upregulation of endogenous angiogenesis, and reductions in lesion areas compared to a phosphate-buffered saline control group [143]. This finding of decreased lesion area was further supported by a subsequent study by Ni et al. [144]. A recent study examining the efficiency of delayed MSC-exosome therapy in TBI found that MSC-exosome administered to Yorkshire swine 9 h post-TBI still demonstrated a significant improvement in neurological recovery rates compared to a normal saline control group [145]. MSC-exosome therapy has also shown promise in modulating microglia activation and neuroinflammation. Several studies have found a significant reduction in microglia polarization and inflammation in MSC-exosome treatment of rodent TBI models [141,144,146].

*5.7. CCR5 Antagonists*

One of the most promising, novel targets of future TBI therapies may be CC chemokine receptor 5 (CCR5). CCR5 is a G-protein-coupled receptor that first gained recognition as an integral coreceptor in HIV cell infection but is now recognized as a significant player in the endogenous activation and trafficking of immune- and oxidative-stress-inducing cells, including macrophages and T cells [147–150]. There is also some evidence that CCR5 may interact with mitochondrial heat shock proteins expressed due to mitochondrial stress and contribute to cell apoptosis [151,152]. Accordingly, CCR5 inhibition has the potential to attenuate some of the effects of mitochondrial stress (Table 1). A recent study by Haruwaka et al. demonstrated, with in vivo imaging during inflammation, that CCR5 performs an integral role in the trafficking of microglia to central nervous system vessels and, consequently, may induce permeability and failure of integrity in the blood–brain barrier (BBB) [153]. These findings indicate that CCR5 may play a role in microglia activation and ROS response post-TBI. Furthermore, there is evidence that CCR5 transcription is upregulated for 7 days following a TBI [154]. This suggests that CCR5 may have a lasting effect post-TBI. Several studies examining TBI outcomes in CCR5 knockout or silenced

rodents compared to WT have consistently demonstrated improved neurological outcomes, reduced fields of damage, and earlier recovery [155–157]. Joy et al. examined outcomes from the Tel Aviv Brain Acute Stroke Cohort study and were able to demonstrate a significant correlation between better stroke outcomes amongst enrollees with a CCR5 loss of function mutation compared to those with CCR5 WT [155]. Potential CCR5 antagonistic therapeutics already FDA-approved for HIV treatment include Cenicriviroc and Maraviroc. Consequently, studies have already demonstrated the effectiveness of Maraviroc as a CCR5 antagonist in rodents with TBI, with outcomes paralleling those found in the knockout studies [155,157].

**Table 1.** Potential treatment options as discussed in Section 5.

| Therapies | Potential Mechanisms of Action |
| --- | --- |
| Immunoglobulin | ↓ p-tau (mitochondrial stress and apoptosis) <br> ↓ caveolin (oxidative stress) |
| Cell-Based | ↓ oxidative stress <br> ↓ inflammatory cell migration |
| MSC-Exosomes | ↓ oxidative stress <br> ↓ ER stress |
| CCR5 Antagonists | ↓ Inflammatory cell migration |
| Extra-synaptic NMDA Receptor Inhibitors | ↓ mitochondrial stress |
| Selective $Ca^{2+}$ Channel Inhibitors | ↓ mitochondrial and ER stress |
| eIF2$\alpha$ Phosphorylation | ↓ unfolded protein production and ER stress |

## 6. Conclusions

Neurotrauma is a leading cause of disability worldwide and can result in secondary sequela with lifelong implications. In this review, mechanisms causing secondary injury in neurotrauma, including ER stress, mitochondrial dysfunction, and oxidative stress, were closely analyzed. In the ER, unfolded pPlroteins aggregate, accumulate, and concomitantly activate the UPR. The UPR under physiologic conditions aims to maintain cellular homeostasis, while the overactivation of it, as seen in traumatic injury, can lead to cell death. Similarly, mitochondrial dysfunction seen in TBI and SCI results in an ATP supply-and-demand mismatch, increased ROS formation, and caspase release through mPTP with resultant apoptosis. Neurotrauma-induced oxidative stress overloads the body's endogenous antioxidant mechanisms and creates a cycle of reactive species formation with ensuing neuroinflammation and apoptosis. Many of these mechanisms are similar to and may kickstart processes similar to those seen in chronic neuroinflammatory and neurodegenerative diseases.

Despite these potentially devasting consequences, there is no FDA-approved treatment for the secondary injury seen in neurotrauma. However, promising treatment options targeting the pathways are emerging. Salubrinal, Guanabenz, and TUDCA target the UPR, while memantine and NIM811 may support normal mitochondrial function. Potential antioxidant therapies include edaravone, Apocyanin/TBHQ, and MitoQ. Other potential therapies are in the form of immunoglobulin, cell-based, MSC-exosome, and CCR5 antagonist therapies. Although showing promising results, many of these remedies are still largely in the pre-clinical phases of investigation. Thus, the continued need to identify treatment options targeting ER stress, mitochondrial dysfunction, and reactive species formation in neurotrauma cannot be understated.

**Author Contributions:** Conceptualization, B.L.-W. and S.A.-M.; writing—original draft preparation, S.A.-M., B.L.-W., J.W., M.W., and E.W.; writing—review and editing, S.A.-M., B.L.-W., and S.Y.; visualization, S.A.-M.; supervision, B.L.-W.; project administration, S.A.-M. and B.L.-W. All authors have read and agreed to the published version of the manuscript.

**Funding:** This research received no external funding.

**Institutional Review Board Statement:** Not applicable.

**Conflicts of Interest:** The authors declare no conflict of interest.

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
