# Peer review of "Secondary Mechanisms of Neurotrauma: A Closer Look at the Evidence"

_diseases, doi:10.3390/diseases10020030_

Round 1

Reviewer 1 Report

The manuscript titled, "Secondary Mechanisms of Neurotrauma: A Closer Look at the Evidence" appears too preliminary. There is lack of in-depth discussion and novelty that would interest the new readers.

Author Response

Thank you for these suggestions. We agree that the manuscript would be strengthened by more in depth discussion and have added several new sections to address. 

Reviewer 2 Report

The review report secondary injury in neurotrauma. However, the relation of neurotrauma and these secondary injury was not explained. Direct link of trauma and these secondary injury must be explained.

Author Response

We appreciate the comments and have focused the paper more specifically to address these key secondary mechanisms throughout. 

Reviewer 3 Report

This review article "Secondary Mechanisms of Neurotrauma: A Closer Look at the 2 Evidence", by Aghili-Mehrizi S et al., reviews the underlying mechanisms of secondary injury in traumatic CNS. They emphasized that oxidative stress and neuroinflammation are responsible for inducing neuronal apoptosis. These events are the beginning moments that last for days to months after the primary injury that led to the secondary injury and proposed potential treatment options.

In this paper, the authors have discussed the underlying mechanisms, summarized the role of antioxidants, and proposed emerging treatment options. Although there is no novelty in this review article, some new information is there. There are many excellent review articles available on this topic.

Author Response

Thank you for these points. We have clarified the timing of oxidative stress further and added several new sections to describe some more novel findings. Your input has helped us strengthen the paper. 

Reviewer 4 Report

The manuscript entitled "Secondary Mechanisms of Neurotrauma: A Closer Look at the Evidence." has been submitted as a review by Aghili-Mehrizi et al.

The manuscript is interesting and contains the relevant information.

Minor Points:

1) Please add a more detailed figure legend to Figure 1. This would support the understanding of this item.

2) Moderate English changes are requested.

Author Response

Thank you for these valuable suggestions. We have strengthened the figure legend and edited the paper throughout. 

Reviewer 5 Report

In this manuscript, the authors aimed to review the impact of neurotrauma at cellular and molecular levels in order to identify potential treatment modalities to secondary injury.

Recently, some other therapeutic approaches, e.g. immunoglobulin, have been proposed. The authors could carefully review the current literature.

The different treatment options (targets, mechanisms of action, published studies and results, etc.) could be briefly summarized in a table.

Author Response

These are excellent points and have been expanded throughout. We added more regarding immunotherapy and the therapeutics in general. The paper is greatly strengthened from the suggestions. 

Round 2

Reviewer 2 Report

The table should fit in the print.

The mechanism of secondary injury caused by direct mechanical trauma is not fully explained.

Author Response

The table formatting has been fixed. Likewise, the direct transition from primary to secondary injury has been better defined in the introduction. 

Thank you for taking time to review and the comments strengthen the manuscript. 

Reviewer 5 Report

Authors replied to my comments. I believe the paper is improved.

Author Response

Thank you for this valuable input and taking time to review our manuscript. We agree and the paper has been strengthened by your suggestions.